# Gene Expression and Chondrogenic Potential of Cartilage Cells: Osteoarthritis Grade Differences

**DOI:** 10.3390/ijms231810610

**Published:** 2022-09-13

**Authors:** Marija Mazor, Eric Lespessailles, Thomas M. Best, Mazen Ali, Hechmi Toumi

**Affiliations:** 1Center for Proteomics, Faculty of Medicine, University of Rijeka, B. Branchetta 20, 51000 Rijeka, Croatia; 2Service de Rhumatologie, Centre Hospitalier Régional d’Orléans, 45000 La Source, France; 3PRIMMO, Plateforme Recherche Innovation Médicale Mutualisée d’Orléans, Centre Hospitalier Régional d’Orléans, 45007 La Source, France; 4Department of Orthopedics, UHealth Sports Medicine Institute, U of Miami, Coral Gables, FL 33146, USA; 5Service Chirurgie Orthopédique et Traumatologique Centre Hospitalier Régional d’Orléans, 45000 La Source, France

**Keywords:** cartilage mesenchymal progenitor cells, osteoarthritis, cartilage repair

## Abstract

Recent data suggest that cells isolated from osteoarthritic (OA) cartilage express mesenchymal progenitor cell (MPC) markers that have the capacity to form hyaline-like cartilage tissue. Whether or not these cells are influenced by the severity of OA remains unexplored. Therefore, we analyzed MPC marker expression and chondrogenetic potential of cells from mild, moderate and severe OA tissue. Human osteoarthritic tibial plateaus were obtained from 25 patients undergoing total knee replacement. Each sample was classified as mild, moderate or severe OA according to OARSI scoring. mRNA expression levels of MPC markers—CD105, CD166, Notch 1, Sox9; mature chondrocyte markers—Aggrecan (Acan), Col II A1, hypertrophic chondrocyte and osteoarthritis-related markers—Col I A1, MMP-13 and ALPL were measured at the tissue level (day 0), after 2 weeks of in vitro expansion (day 14) and following chondrogenic in vitro re-differentiation (day 35). Pellet matrix composition after in vitro chondrogenesis of different OA-derived cells was tested for proteoglycans, collagen II and I by safranin O and immunofluorescence staining. Multiple MPC markers were found in OA cartilage resident tissue within a single OA joint with no significant difference between grades except for Notch1, which was higher in severe OA tissues. Expression levels of CD105 and Notch 1 were comparable between OA cartilage-derived cells of different disease grades and bone marrow mesenchymal stem cell (BM-MSC) line (healthy control). However, the MPC marker Sox 9 was conserved after in vitro expansion and significantly higher in OA cartilage-derived cells compared to its levels in the BM-MSC. The in vitro expansion of cartilage-derived cells resulted in enrichment while re–differentiation in reduction of MPC markers for all three analyzed grades. However, only moderate OA-derived cells after the in vitro chondrogenesis resulted in the formation of hyaline cartilage-like tissue. The latter tissue samples were also highly positive for collagen II and proteoglycans with no expression of osteoarthritis-related markers (collagen I, ALPL and MMP13). MPC marker expression did not differ between OA grades at the tissue level. Interestingly after in vitro re-differentiation, only moderate OA-derived cells showed the capacity to form hyaline cartilage-like tissue. These findings may have implications for clinical practice to understand the intrinsic repair capacity of articular cartilage in OA tissues and raises the possibility of these progenitor cells as a candidate for articular cartilage repair.

## 1. Introduction

Articular cartilage is an avascular and aneural tissue with little intrinsic capacity for self-repair [1,2]. Accordingly, injury to articular cartilage often leads to progressive tissue degradation and ultimately degenerative joint disease or osteoarthritis (OA) [2]. Currently, autologous chondrocyte transplantation (ACT) is one of the most advanced methods for cartilage repair [3,4]. ACT is a cell-based therapy involving chondrocyte isolation from preserved areas of the OA-affected cartilage, in vitro expansion, and re-implantation into the affected zone [3]. Unfortunately, biopsy of the preserved cartilage frequently leads to donor site morbidity. In addition, isolated chondrocytes exhibit low proliferation capacity and show an inability to retain chondrogenic potential after in vitro expansion [4].

It is well recognized that mesenchymal stem/progenitor cells (MSC/MPC) have a robust clonal self-renewal and multilineage differentiation potential suggesting an alternative option for the treatment of articular cartilage lesions. Mesenchymal progenitor cells (MPC) have been successfully isolated from mature human cartilage [5,6,7,8,9,10]. Stoop and colleagues demonstrated that OA-derived MPC cells were capable of producing cartilage-like tissue after injection in an in vivo mouse model [11]. When compared to bone marrow MSC (BM-MSC), OA cartilage cells showed higher cartilaginous matrix deposition following in vitro chondrogenesis [12]. In fact, chondrogenesis of BM-MSC resulted largely in fibrocartilaginous tissue production followed by expression of hypertrophic markers, Col X and ALPL [12].

While BM-MSC are accepted sources for stem cell-based cartilage repair therapy, MPC from patients with OA are a new and potentially attractive option [13]. In fact, resident cells in the diseased cartilage tissue may be ideal for cartilage repair because they are already active in the host tissue and potentially safer than exogenous cells [14]. Recently, progress has been made in understanding the pathogenesis of OA and the involvement of these progenitor cells. Several research groups have reported observations on progenitor cells from human articular cartilage tissue [7,10,15,16,17,18,19]. However, the results regarding appearance and prevalence of MPC in normal versus OA cartilage are still controversial [5,6,7,15,19]. For instance, Alsalameh et al. (2004) indicated that multipotent MPCs are present in adult human articular cartilage and that their presence is increased in OA cartilage [6]. It has been reported that the proportion of CD105+CD166+ cells in OA cartilage is ~8%, compared with ~4% in normal cartilage [6]. Nevertheless, others reported no OA grade-specific differences in stem cell marker expression [7]. In fact, to date, studies have often been limited in comparing OA-affected cartilage to normal cartilage despite the fact that OA is typically a progressive joint disease that does not uniformly involve the entire joint. Whether the MPC population and the degree to which their chondrogenic potential are preserved across the disease spectrum remains incompletely understood [17]. Accordingly, we therefore sought to examine changes in MPC, chondrocytes and osteoarthritis-related markers for different OARSI (Osteoarthritis Research Society International) grades and the potential of cells derived from different OA grades to form hyaline-like cartilage. Understanding the development and function of MPC in vivo at different stages of OA may be important to the design of MPC-based OA therapies.

## 2. Results

The experimental protocol procedure is shown in Figure 1. Forty-four samples were collected from 25 subjects and placed in one of three groups based on the Osteoarthritis Research Society International (OARSI) scoring system described below: mild OA (grade 1, n = 7), moderate OA (grade 2, n =18) and severe OA (grade 3 and 4, n =19). mRNA levels of MPC markers were measured in the tissue isolated from each grade (the complete analysis performed on tissue has been marked as D0 in the following text). The mRNA levels of MPC, chondrocytes and hypertrophic chondrocytes markers were measured in cells isolated from each grade after being expanded in vitro for 14 days (D14). Furthermore, evolution of MPC, chondrocytes and hypertrophic chondrocytes markers between D0, D14 and D35 (after re-differentiation of expanded cells) were recorded. Cartilage matrix formation at D35 was confirmed with Safranin O staining for proteoglycans and immunostaining for collagen II and I. The most significant findings are presented below. 

### 2.1. mRNA Expression Profile of MPC Markers in Tissue Isolated from Mild, Moderate and severe OA Grade

To characterize changes in the expression of the MPC (CD105, CD166, Notch-1 and Sox-9) markers associated with different OA grades, a qPCR analysis was performed on tissue (D0) isolated for all 3 OA grades (mild, moderate and severe). There was no significant difference in the expression of CD105, CD 166, and Sox 9 markers between grades (Figure 2). Notch-1 expression was significantly higher in severe compared to moderate OA tissue samples (Figure 2).

### 2.2. Cell Number, Viability, Morphology and Proliferation Rate of Cells Derived from Mild, Moderate and Severe OA Grade

Our results did not show any significant difference in cell number between the three OA grades (Figure 3A, left panel). Similarly, cell viability determined by the live/dead ratio after the digestion step did not show any significant grade differences (Figure 3A, middle panel). Cell proliferation capacity measured as the variation in total cell number before and after 14 days of in vitro expansion demonstrated no significant difference between the grades (Figure 3A, right panel). Cell morphology revealed the presence of spindle-shaped, fibroblast-like cells for all three grades (Figure 3B).

### 2.3. Expression Levels of MPC, Chondrocyte and Hypertrophic Chondroyte Markers after In Vitro Expansion of Cells Derived from Mild, Moderate and Severe OA Grades

In order to explore the chondroprogenitor properties after in vitro expansion, we analyzed MPC, chondrocyte and hypertrophic chondrocyte marker expression in cells derived from the different OA grades and compared it to the BM-MSC line. This comparative analysis showed no prevalence of CD 105 in the BM-MSC compared to cells derived from mild and moderate OA specimens (Figure 4A). CD105 mRNA expression was significantly higher in BM-MSC compared to severe OA-derived cells (Figure 4A). Moreover, a higher level of this marker was observed in moderate compared to severe OA-derived cells. The CD166 was significantly higher in the BM-MSC compared to OA cells for all three grades (Figure 4A). Notch-1 m RNA was significantly up-regulated only in the moderate OA-derived cells when compared to its levels in BM-MSC (Figure 4A). The MPC marker Sox 9 and chondrocyte markers Col II and Acan were significantly higher in cells derived from OA tissues compared to the BM-MSC (Figure 4B). Sox9 mRNA expression was also significantly higher in mild and moderate than in severe OA-derived cells (Figure 4B). Acan mRNA expression levels after in vitro expansion were higher in moderate compared to severe OA-derived cells (Figure 4B). The hypertrophic marker Col I did not differ between BM-MSC and cells derived from OA cartilage (Figure 4C). Hypertrophicmarkers (MMP13 and ALPL) were both higher in OA cartilage-derived cells compared to the BM-MSC (Figure 4C). The *p*-values are presented in Appendix A.

### 2.4. The Changes in MPC, Chondrocyte and Hypertrophic Chondroyte Marker Expression between Tissue (D0), Expanded (D14) and Re-Differentiated (D35) Cells Derived from Mild, Moderate and Severe OA

Mild OA-derived cells: There was a significant increase of CD105, CD166 and Notch 1 between D0 and D14 (Figure 5A). The mRNA levels of these MPC markers significantly declined at D35 compared to D14 (Figure 5A). These results were similar for Sox 9, an MPC marker significantly expressed in the early stage of differentiation from MSC to chondrocytes (chondrogenesis) (Figure 5B). Yet, markers for mature chondrocytes, Acan and Col II A 1 did not significantly increase at D35 compared to D14 (Figure 5B). Acan and Col II expression decreased in D35 compared to D0 and D14 specimens. There was no significant difference in Col I A1, a marker of hypertrophic chondrocytes, at D0, D14 and D35 (Figure 5B). The *p*-values are presented in Appendix A.

The expression of MPC, chondrocyte and hypertrophic chondrocyte markers are presented in Figure 6 and Figure 7. Moderate OA-derived cells: MPC marker expression at D14 compared to D0 was similar to that in mild OA. CD105, CD166 and Notch-1 significantly increased at D14 compared to D0 (Figure 6A). In line with mild OA, CD105 and Notch 1 significantly decreased at D35 compared to D14. In contrast to mild OA cells, in moderate OA, CD166 increased between D14 and D35 as well as between D0 and D35 (Figure 6A). Sox 9 did not show any significant change between D0 and D14, however, it significantly decreased at D35 compared to D14 (Figure 6B). Similarly, to mild OA, moderate OA cells expressed significantly lower levels of Acan and Col II A 1 at D14 compared to D0 (Figure 6B). Acan expression was also decreased at D35 compared to D14 (Figure 6B). Interestingly, as shown above, mild OA Col II A1 significantly decreased between D14 and D35 (Figure 5B), while in moderate OA, this chondrocyte marker significantly increased at D35 compared to D14 (Figure 6B). Moreover, Col I A1, a marker of hypertrophic chondrocytes, showed a significant decrease at D35 compared to D14 and D0 (Figure 6B), which was not the case in mild OA (Figure 5B). In addition, ALPL and MMP-13 markers were significantly down-regulated only in moderate OA-derived pellets after in vitro chondrogenesis (Figure 8D). In contrast, chondrogenesis of the cells derived from severe OA specimens demonstrated a significant increase of the catabolic marker MMP13 (Figure 8D). The *p*-values are presented in Appendix A.

Severe OA-derived cells: Similarly to the other two grades, CD105 and CD166 marker expression significantly increased from D0 to D14 (Figure 7A). There was no significant difference for Notch 1 (Figure 7A). CD105 and Notch 1 significantly decreased at D35 compared to D14, while CD166 significantly increased (Figure 7A). Sox 9 expression was decreased at D14 compared to D0 (Figure 7B) and was significantly lower at D35 compared to D0 (Figure 7B). mRNA levels of Acan significantly decreased from D0 to D14, and D14 to D35 (Figure 7B). Col II A1 significantly decreased from D0 to D14 and remained stable between D14 and D35. There was no significant difference in Col I A1 marker expression between D0, D14 and D35 (Figure 7B). The *p*-values are presented in Appendix A.

### 2.5. The Potential of Mild, Moderate and Severe OA-Derived Cells to Form Hyaline Cartilage-Like Tissue In Vitro

Histological staining (safranin O/hematoxylin/fast green) demonstrated significant proteoglycan deposition in moderate ((Figure 8A; middle panel) compared to mild (Figure 8A; left panel) and severe (Figure 8A; right panel) OA-derived pellets. Moreover, the main cartilage marker collagen II stained highly positive in moderate (Figure 8B; middle panel) compared to the other two grades of OA (Figure 8B; left and right panel). Conversely, the marker of fibrocalcified cartilage, collagen I, showed highly positive staining in severe OA (Figure 8C; right panel) compared to moderate (Figure 8C; middle panel) and mild (Figure 8C; left panel) specimens. The negative control stained only with the secondary antibody (no primary antibody) is presented in the top right corner of each IHC image. Moreover, ALPL and MMP-13 hypertrophic chondrocyte markers were significantly down-regulated only in moderate OA (Figure 8D; middle panel)-derived pellets after in vitro chondrogenesis. In contrast, chondrogenesis of the cells-derived from severe OA resulted in a significant increase of the catabolic marker MMP13 (Figure 8D; right panel). 

## 3. Discussion

Cell-surface markers identified presence of multipotent MPC subsets in both healthy and OA-affected cartilage. It has been shown that MPC compared to mesenchymal stem cells from other sources have a higher potential for cartilage repair [20,21]. It is felt that these cells already possess the developmental repertoire of native tissue that may positively impact the formation of neo-cartilage. The higher prevalence of MPC in the OA-affected cartilage samples compared to healthy cartilage has been confirmed previously, suggesting their potential importance in joint repair [5,6,19]. However, it remains controversial how the MPC markers and chondrogenic potential vary with the degree of OA. To date, studies have mostly compared MPC isolated from OA cartilage without dividing it into OA grades to tissue specimens isolated from healthy cartilage [5,6,19]. Herein, we have identified and separated different OA grades based on histology (OARSI scoring) and compared their MPC marker expression and chondrogenic potential. We measured mRNA levels of MPC-related markers CD105, CD166, Notch-1 and Sox9 and chondrocyte/hypertrophic chondrocyte markers Col II, Aggrecan, Col I, ALPL, MMP13 at the tissue level after two weeks of two-dimensional in vitro expansion and three weeks of 3D re-differentiation. The formation of hyaline cartilage-like tissue in 3D pellets derived from different OA grades was further tested at the protein level to more fully characterize their potential for chondrogenesis. 

Our findings indicated that MPC markers are present in articular cartilage from OA adult human tissues and that their expression does not appear to be highly influenced by the degree of OA. However, after in vitro expansion and 3 D in vitro re-differentiation to hyaline cartilage tissue only moderate OA-derived cells resulted in hyaline cartilage-like tissue with no signs of fibrocartilagenous tissue. In contrast, pellets derived from severe OA specimens demonstrated strong collagen I staining, which eliminates it as a reliable source for cell based cartilage repair. Thus, moderate OA-derived cells may be an interesting target for cell-based regenerative therapies in degenerative joint diseases such as OA. 

Our initial results showed no significant difference in MPC mRNA marker expression levels between tissues isolated for different OARSI-based OA grades. Previous studies demonstrated that in vitro expansion enriches a subpopulation of cells initially positive for MPC markers [5,6,10]. In line with these findings, we showed that mRNA expression of MPC markers generally increased after 2 weeks of in vitro expansion across all three OA grades with no significant differences between grades. Our results confirm those previously reported by Bernstein et al. [7]. 

Despite the role of notch signaling in differentiation, proliferation and apoptosis, its role in OA remains unknown [22]. Studies on the interface between Notch signaling and cartilage sub-populations have shown that the expression of Notch-1 receptor and mesenchymal progenitor populations are associated [23]. Namely, several studies identified Notch-1 as a marker of MPC in cartilage in both normal and OA tissue. In addition, Dehne et al., 2009 and Grogan et al., 2009 reported that the frequency of cells expressing Notch-1 is higher in damaged OA cartilage compared to healthy tissue [19,24]. Based on these previous reports, we used Notch-1 as a marker of MPCs. Our findings demonstrated an increase of Notch 1 in mild and moderate tissues, while no change was observed in severe OA samples following 2 weeks of in vitro expansion. Yet, we demonstrated that the tissue levels of these markers were already high in severe OA samples. In line with these results, previous reports demonstrated that sustained Notch signaling in joint cartilage leads to progressive OA pathological changes that could explain its high expression in severe OA specimens. Taken together, the high levels of these markers in severe OA tissue potentially contribute to OA progression by promoting the expression of cartilage-related proteases (MMPs and ADAMTSs), collagen type I and inflammatory factors, as reported by Liu et al., while transient in vitro upregulation of this marker in mild and moderate OA may promote its enrichment in MPC and consequently increase cartilage extracellular matrix (ECM) synthesis [25]. 

In our study, Sox 9 mRNA significantly increased between D0 and D14 in mild OA tissues while decreasing in moderate and severe specimens. It has been shown that Notch-1 is one of the triggers of MSC differentiation and Sox 9 is a transcription factor that induces MSC chondrogenesis as well as expression of chondrocyte-specific genes such as collagen II and aggrecan [26,27]. Therefore, the general increase of CD105, CD166, Sox 9 and Notch-1 with the expansion of cells derived from mild and moderate OA cartilage could suggest that these cells potentially have higher differentiation and regeneration capacity than severe OA cartilage cells.

ACAN mRNA (a marker of mature chondrocytes) significantly decreased at D35 compared to D14 for all three OA grades. This result at first implicates that OA cartilage –-derived cells were not able to differentiate into mature chondrocytes. Yet, histological analysis of pellets revealed positive proteoglycan staining at D35, particularly in the moderate OA-derived pellets. This is in accordance with a recent study showing the highest mRNA expression of ACAN during the first week of chondrogenesis, followed by a decline in the subsequent 2 weeks of differentiation [28]. In contrast, Xu et al. demonstrated the highest mRNA ACAN expression between 18–24 days of chondrogenesis [29]. Taken together, these findings require further ACAN mRNA analysis at different time points of chondrogenesis. Nevertheless, Mwale et al. concluded that mRNA ACAN levels may not be the best indicator of chondrogenesis, as they did observe relatively high expression of this marker even in BM-MSC [30].

The major objective of the present study was to investigate the composition of in vitro obtained cartilage tissue after the re-differentiation of cells isolated from different OA grades. Our approach demonstrated higher proteoglycan deposition in moderate derived pellets compared to mild and severe OA specimens. Moreover, the immunofluorescence of hyaline-like cartilage marker collagen II showed high positive staining in moderate OA-derived pellets in comparison to the other two grades, where staining was negligible. However, it is important to note that no hypertrophic cartilage markers, including collagen I, ALPL and MMP13 were observed in moderate-OA-derived pellets. Interestingly, severe OA-derived pellets showed positive staining of collagen I confirmed by immunofluorescence and a significant increase in both MMPl3 and ALPL mRNA after 3D differentiation of expanded cells. Brew and colleagues compared affected and unaffected areas of OA cartilage and found no difference in neo-cartilage tissue derived from cells isolated from these two sources [31]. Notably, they combined all tissues from different grades. Our approach to separate tissues from different grades permitted us to highlight that the neo-cartilage derived from moderate OA specimens had a greater potential to form hyaline-like tissue compared to cells obtained from mild and severe OA specimens. 

## 4. Materials and Methods

### 4.1. Cartilage Preparation

Human osteoarthritic (OA) tibia plateaus were obtained from the knee joints of 25 patients (58 to 85 years; mean 68 years) with OA who elected to undergo total joint arthroplasty. Patients with systemic inflammatory diseases such as rheumatoid arthritis or spondyloarthropathies were excluded. The protocol was approved by the CPP (Comité de Protection des Personnes Nord Ouest IV, Protocole RT-07, biological collection: CB.2012. RT.07) and patients gave their consent for the use of samples for research.

Several zones of OA tibial plateaus from all 25 patients were macroscopically graded by two independent observers according to the Outerbridge scale (Grade 0: intact cartilage; Grade 1: minimal fibrillation and softening; Grade 2: partial-thickness defect with fissures on the surface; Grade 3: fissuring to the level of subchondral bone; Grade 4: total loss of cartilage and exposure of subchondral bone). Each macroscopically scored OA grade was further divided into three portions. One portion was used for the isolation of the cartilage tissue (D0) on which we examined the expression of MPC markers. The second portion was used to isolate the tissue grafts encompassed cartilage and subchondral bone; these grafts were used to histologically examine level of cartilage degradation by OARSI scoring. Finally, the third portion was used to isolate cells from cartilage tissue which were then expanded in vitro for 2 weeks (D14), one part of the expanded cells was used for the q-PCR analysis of MPC and chondrocyte/hypertrophic chondrocyte markers expression and one part for the in vitro re-differentiation into mature chondrocytes (D35). The potential of cartilage cells isolated from different grades to re-differentiate to mature chondrocytes after in vitro expansion and to form hyaline-like cartilage tissue was analyzed by q-PCR, standard histology (HES), and immunohistochemistry for specific hyaline cartilage and hypertrophic cartilage markers. 

### 4.2. Histological Assessment of OA Grafts

Eight mm diameter grafts were isolated from OA samples and fixed in paraformaldehyde 4% (Sigma-Aldrich, Lyon, France), decalcified in EDTA 15% (Sigma-Aldrich, Lyon, France) for 3 weeks and embedded in paraffin. Sections of 5 µm diameter were stained with hematoxylin–eosin–saffron (HES) (Sigma-Aldrich, Lyon, France) staining and scored by two blinded observers according to the OARSI grading system (Grade 0: intact cartilage surface, normal matrix morphology and intact cells orientation; Grade 1: superficial fibrillation and cells death and hypertrophy; Grade 2: matrix depletion in the superficial and middle zone and disorientation of chondrocyte columns; Grade 3: matrix vertical fissures into mid-zone and formation of the chondrocyte clusters; Grade 4: loss of superficial and mid-zone of cartilage, Grade 5 and 6: no cartilage left and bone deformation is present) [32].

### 4.3. Isolation and Culture of Cells from Cartilage Samples

For cell isolation, cartilage slides for each grade were harvested with a stainless-steel scalpel and washed twice in PBS supplemented with Penicillin/Streptomycin (PS) 1% (Gibco™, Courtaboeuf, France). Slides were then incubated for 4 hours at 37 °C in serum-free DMEM/F12 (Gibco™, Courtaboeuf, France) containing collagenase B 0.2% (Roche, Mannheim, Germany). Supernatant was then collected and centrifuged under 1500 rpm to pellet cells. Cells were washed once in PBS/1% PS. A second enzymatic digestion for 3 hours was performed on the remaining undigested tissue. Cells were washed once in PBS/1% PS, pooled with cells from the first digestion, counted and seeded in DMEM/F12 supplemented with FBS 10%, PS 1% and Fungizon 0.1% (Gibco™, Courtaboeuf, France). Cells were cultured for 2 weeks in DMEM/F12 supplemented with FBS 10%, PS 1% and Fungizon 0.1%. In the 2-week period of time, media was changed three times a week and cells were dissociated by trypsin treatment (Gibco™, Courtaboeuf, France) when they reached 80% of confluence. The expanded cells were used for qPCR analysis and differentiation assays.

### 4.4. Cell Differentiation Assay

At day 14, expanded cells were pelleted at a concentration of 2.5 × 10^5^/0.25 mL for qPCR analysis and 1 × 10^6^/2.5 mL for histology and immunostaining analysis. Cells were centrifuged at 2000 rpm for 5 min in 1.5 mL Eppendorf tubes to form spherical pellets. To stimulate chondrogenic differentiation, pellets were cultured in DMEM/F12 media supplemented with ITS 1% (Insulin-Transferrin-Selenium, Gibco, France), 10^−4^ mM Dexamethasone, 1 mM Sodium-pyruvate, 0.15 mM L-ascorbic acid, 0.35 mM Proline, 10 ng/mL TGF-β, 1% PS and 10% FBS. Pellets were then cultured for 21 days with a medium change every other day. Fifteen samples were tested with the differentiation assay.

### 4.5. mRNA Expression/Quantitative Real-Time PCR (qPCR)

Total RNA was extracted from BM-MSC cell line (StemPro™ BM Mesenchymal Stem Cells, GibcoTM, Thermo Fisher Scientific, Cat No. A15652, Waltham, MA, USA), cartilage tissue (D0), 2-week expanded (D14) and 3-week re-differentiated cartilage cells (D35) with the RNeasy mini kit (Qiagen, Paris, France) according to the manufacturer’s instructions. Tissue homogenization was performed in liquid nitrogen with a handheld homogenizer (ProScientific, USA). CDNA was obtained with the Quanti Tect Reverse Transcription kit according to the manufacturer’s instructions (Qiagen, Paris, France). QPCR amplification was carried out with the CFX96 Real-Time System (Bio-Rad, Marnes-la-Coquette, France). Cells and tissue were tested for expression of the following primers: CD105 (Cat. No QT00013335, Qiagen, Paris, France); CD166 (Cat. No Q100026824, Qiagen, Paris, France); Notch 1 (Cat. No QT00001365, Qiagen, Paris, France); ACAN (Cat. No QT00001365, Qiagen, Paris, France); Col II A1 (Cat. No. QT00049518, Qiagen, Paris, France); Col I A2 (Cat. No. QT0007, Qiagen, Paris, France); and MMP-13 (Cat. No QT00001764, Qiagen, Paris, France). Sox9 (5’GCTCTGGAGACTTCTGAACGA3’, 5’GGGAGATGTGCGTCTGCT3’) and ALPL (5’TGCACCATGATTTCACCATT3’, 5’CGTTGGTGTTGAGCTTCTGA3’) primers were synthesized by Invitrogen^TM^ (ThermoFisher Scientific, Courtaboeuf, France). Expression of the housekeeping gene GAPDH was used to normalize gene expression (Cat. No QT01192646, Qiagen, Paris, France). The 2-ΔΔCt method was used for relative gene expression analysis.

### 4.6. Histology and Immunostaining

To visualize the cell structure, 2-week expanded cells (D14) were subjected to phaloidine staining of F-actine (Invitrogen, Alexa Fluor™ 594 Phalloidin, Cat. No: A12381). Before staining, cells were fixed (4% paraformaldehyde/20 min/RT), permeabilized (PBS-Triton 0.1%-Triton 100x; Acros organics, Geel, Belgium), and non-specific sites were blocked with 1% bovine serum albumin for 30 minutes. At day 35, cell pellets were fixed in paraformaldehyde 4% (Sigma-Aldrich, Lyon, France), embedded in paraffin, and cross-sectioned at 5 µm. Samples were used for standard safranin O/hematoxylin/fast green staining (Sigma-Aldrich, Lyon, France) and for immunofluorescence staining of the cartilage component-collagen II and fibrocartilage component-collagens I. To unmask antigens of interest, the sections immune-stained with collagen II were pre-treated with proteinase K, while sections stained with collagen I with citric buffer for 25 min. All sections were then permeabilized (PBS-Triton 0.1%-Triton 100x; Acros organics, Geel, Belgium), blocked with 1% bovine serum albumin for 1 hour, and incubated with primary antibodies overnight at 4 °C in a humid chamber. After the primary antibody staining, sections were incubated for 1hour at room temperature with the 549-conjugated anti-rabbit (1:1500; Rochland, Limerick, Ireland). Control sections were treated only with the secondary antibody and served as a negative control. Cell nuclei were stained with DAPI (4’, 6-diamidino-2-phenylindole) and samples were mounted in immunostaining mounting solution (ProLong™ Diamond Antifade Mountant with DAPI P36962, Thermo Fisher Scientific, Waltham, MA, USA). Fluorescence-labeled sections were visualized with a BA400 microscope (Motic^®^ Wetzlar, Wetzlar, Germany). Pellet sections incubated only with the-secondary antibodies showed no staining.

### 4.7. Statistical Analysis

Data are reported as median with 95% confidence intervals. For comparative analysis of the three groups, results were analyzed with the non-parametric Kruskal–Wallis test. A non-parametric Mann–Whitney test was used to evaluate the significance. The critical *p*-value for statistical significance was *p* = 0.05.

## 5. Conclusions

Our findings indicate that multipotent MPCs are present in adult human OA articular cartilage tissues and that their frequency is not affected by the severity of OA as defined by the OARSI scoring system. Interestingly, after in vitro re-differentiation only moderate OA-derived cells showed the capacity to form hyaline cartilage-like tissue. The use of a resident cartilage progenitor cell population for the treatment of cartilage defects by employing tissue engineering principles is, therefore, suggested by our results. This observation may have an implication for understanding the intrinsic repair capacity of articular cartilage and to monitor when and which MPCs are suitable for cartilage repair strategies. Further research with a particular focus on moderate OA tissues appears warranted.

## Figures and Tables

**Figure 1 ijms-23-10610-f001:**
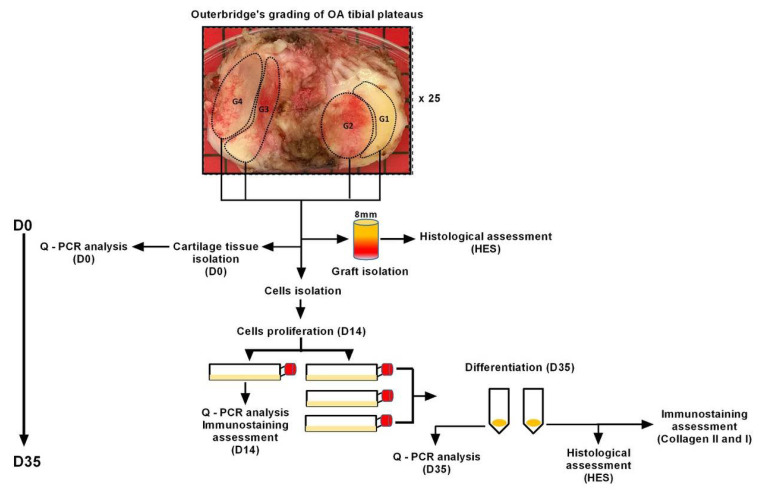
Schematic illustration of experimental design. First the macroscopic Outerbridge classification of OA was used to divide the specimens into different zones. (Grade 1/G1: minimal cartilage degradation; Grade 2/G2: partial-thickness defect with fissures on the surface; Grade 3/G3: fissures to the level of subchondral bone and Grade 4/G3: loss of cartilage and exposure of subchondral bone). Each grade was further subdivided into three portions; portion 1 for isolation of the cartilage tissue (D0-qPCR), portion 2 for histology and OARSI grade examination and portion 3 for isolation of cells from cartilage tissue, their expansion (D14-qPCR) and re-differentiation (D35-qPCR, histology and immunostaining).

**Figure 2 ijms-23-10610-f002:**
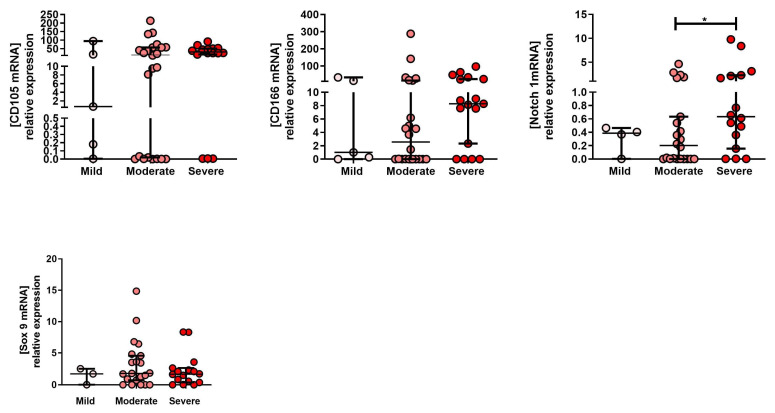
MPC marker expression in different OA grades. The mRNA expression levels of CD105, CD166 and Notch 1 (MPC markers) in tissue (D0) isolated from mild, moderate and severe OA grades. * *p* < 0.05, non-parametric Kruskal–Wallis and Mann–Whitney test.

**Figure 3 ijms-23-10610-f003:**
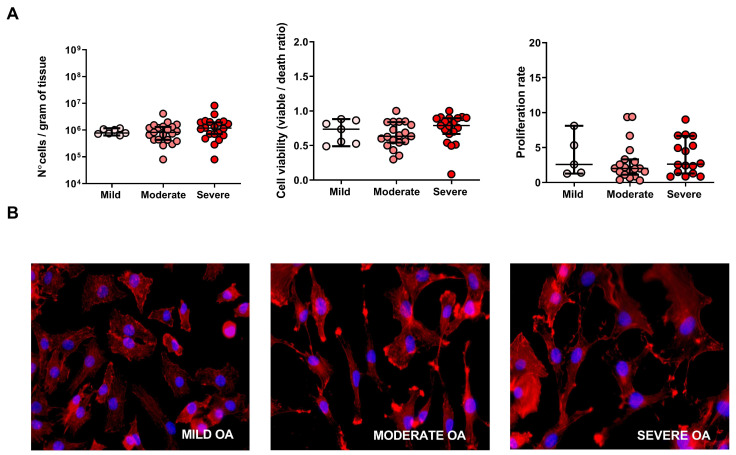
Cell number, viability and proliferation rate of cells derived from mild, moderate and severe OA regions. (**A**) The number of cells obtained from different OA grades after tissue digestion (**left** panel). The ratio of live/death cells after the isolation from the different OA grades (**middle** panel). The ratio between cell number obtained after the isolation from tissue and after the 2 weeks of in vitro proliferation (D14) (**right** panel). (**B**) Immunofluorescence of F-actin by phaloidine to visualize the morphology of cells derived from mild, moderate and sever OA. A non-parametric Kruskal–Wallis and Mann–Whitney test.

**Figure 4 ijms-23-10610-f004:**
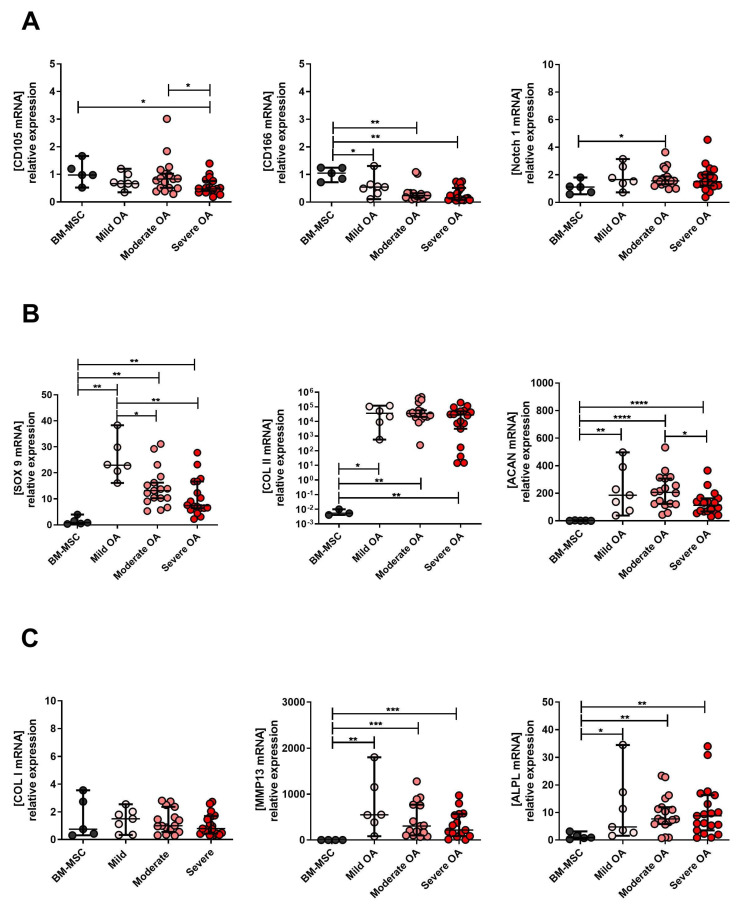
The mRNA expression of MPC, chondrocyte and hypertrophic chondrocyte markers in mild, moderate and severe OA-derived cells compared to BM-MSC. (**A**) mRNA expression levels of CD105, CD166 and Notch 1 (MPC markers) in BM-MSC cell line and mild, moderate and severe OA-derived cells expanded in vitro for 2 weeks (D14). (**B**) mRNA expression levels of Sox9 (MPC marker), Acan, Col II (chondrocytes markers) in mild, moderate and severe OA-derived cells expanded for 2 weeks in vitro (D14) and BM-MSC cell line. (**C**) The mRNA expression levels of Col I, MMP13 and ALPL (hypertrophic chondrocytes markers) in mild, moderate and severe OA-derived cells expanded for 2 weeks in vitro (D14) and BM-MSC cell line. * *p* < 0.05, ** *p* < 0.01, *** *p* < 0.001, **** *p* < 0.0001, non-parametric Kruskal–Wallis and Mann–Whitney test.

**Figure 5 ijms-23-10610-f005:**
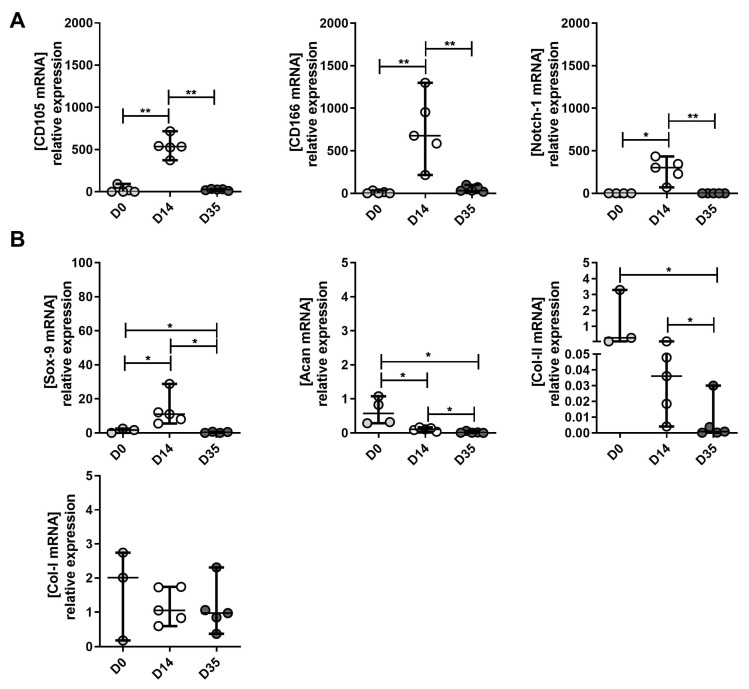
The expression of MPC, chondrocyte and hypertrophic chondrocyte markers between mild OA-derived tissue (D0), expanded (D14) and re-differentiated cells (D35). (**A**) Changes in mRNA expression levels of CD105, CD166 and Notch 1 (MPC markers) between D0, D14 and D35. (**B**) The changes of mRNA expression levels of Sox9 (MPC marker), Acan and Col II (chondrocytes markers) and Col I (hypertrophic chondrocytes marker) between D0, D14 and D35. D0 is used as the control to normalize expression levels. * *p* < 0.05, ** *p* < 0.01, non-parametric Kruskal–Wallis and Mann–Whitney test were used.

**Figure 6 ijms-23-10610-f006:**
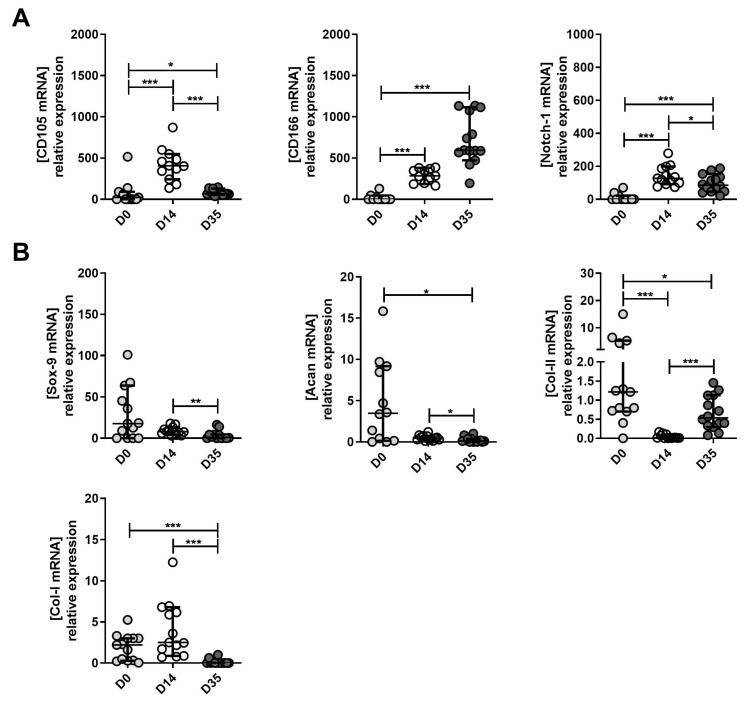
The expression of MPC, chondrocyte and hypertrophic chondrocyte markers between moderate OA-derived tissue (D0), expanded (D14) and re-differentiated cells (D35). (**A**) Changes in mRNA expression levels of CD105, CD166 and Notch 1 (MPC markers) in D0, D14 and D35. (**B**) The changes in mRNA expression levels of Sox9 (MPC marker), Acan and Col II (chondrocytes markers) and Col I (hypertrophic chondrocytes marker) between D0, D14 and D35. D0 is used as the control to normalize expression levels. * *p* < 0.05, ** *p* < 0.01, *** *p* < 0.001, non-parametric Kruskal–Wallis and Mann–Whitney test were used.

**Figure 7 ijms-23-10610-f007:**
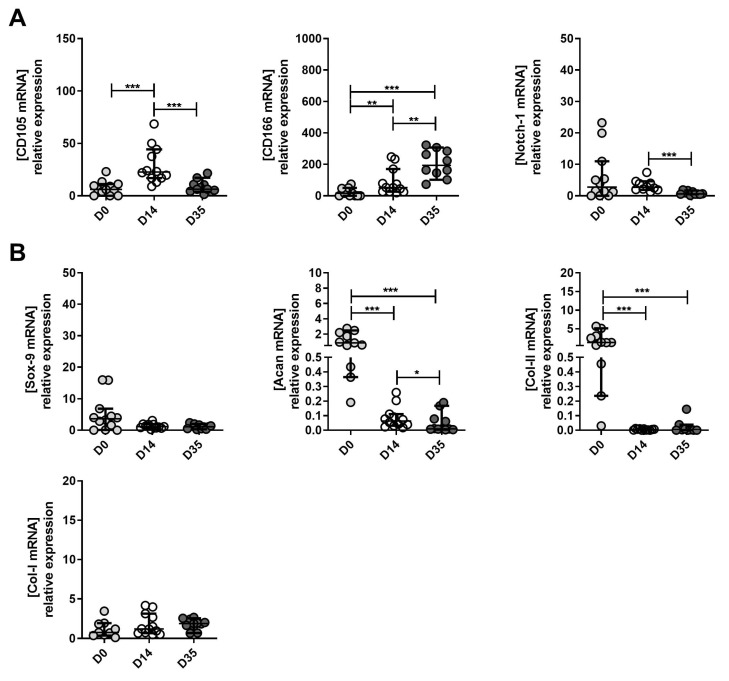
The expression of MPC, chondrocyte and hypertrophic chondrocyte markers between severe OA-derived tissue (D0), expanded (D14) and re-differentiated cells (D35). (**A**) Changes in mRNA expression levels of CD105, CD166 and Notch 1 (MPC markers) between D0, D14 and D35. (**B**) The changes of mRNA expression levels of Sox9 (MPC marker), Acan and Col II (chondrocytes markers), and Col I (hypertrophic chondrocytes marker) between D0, D14 and D35. D0 is used as the control to normalize expression levels. * *p* < 0.05, ** *p* < 0.01, *** *p* < 0.001, non-parametric Kruskal–Wallis and Mann–Whitney test.

**Figure 8 ijms-23-10610-f008:**
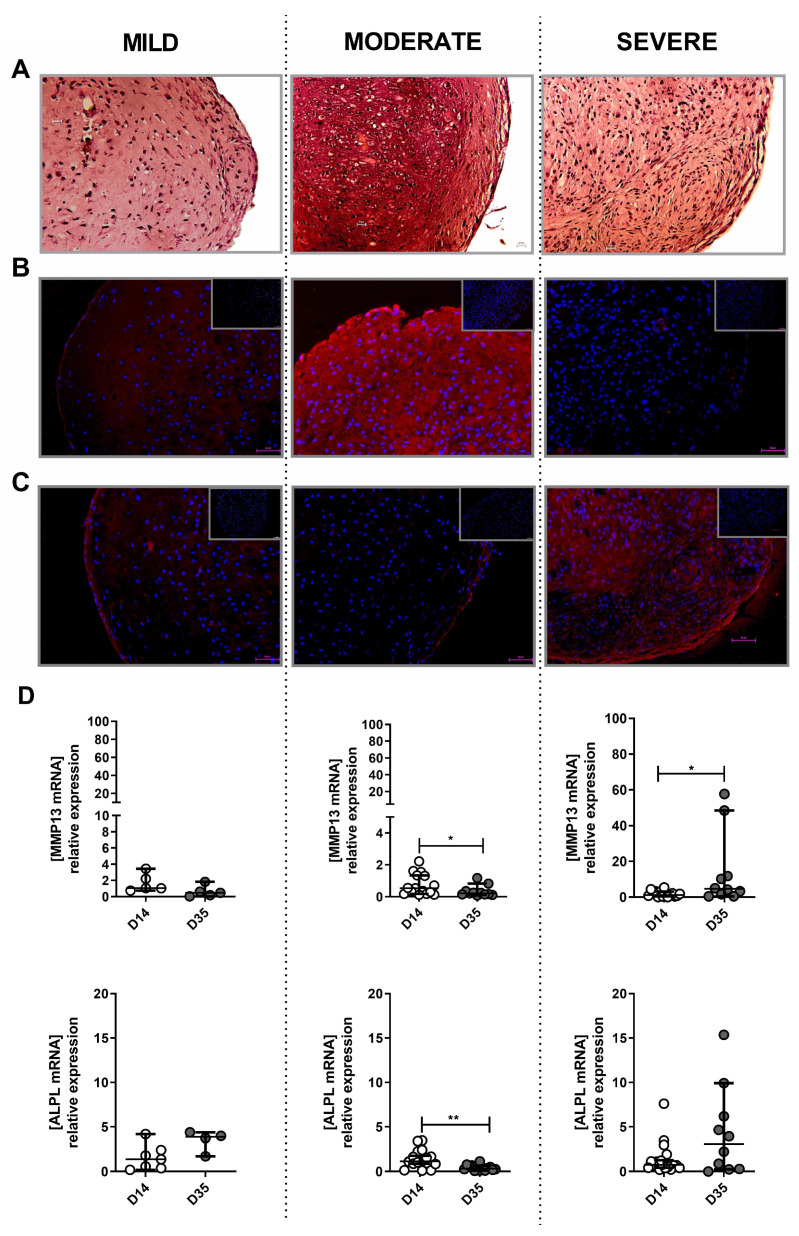
HES, Collagen II and I immunostaining of pellets of re-differentiated cells derived from mild, moderate and severe OA specimens. (**A**) Representative images of safranin O/hematoxylin/fast green of pellet sections derived from mild (left panel), moderate (middle panel) and severe (right panel) OA. (**B**) Representative images of immunofluorescence of Collagen II on pellet sections derived from mild (left panel), moderate (middle panel) and severe (right panel) OA samples. (**C**) Representative images of immunofluorescence of Collagen I on pellet sections derived from mild (left panel), moderate (middle panel) and severe OA tissues (right panel). Representative images of negative control (without primary antibody) are presented in the right corner of each IHC representative image. (**D**) Changes in MMP13 and ALPL mRNA expression levels between D14 and D35 in mild (left panel), moderate (middle panel) and severe OA (right panel) derived cells. * *p* < 0.05, ** *p* < 0.01, non-parametric Kruskal–Wallis and Mann–Whitney test were used.

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
