# Peer review of "Gene Expression and Chondrogenic Potential of Cartilage Cells: Osteoarthritis Grade Differences"

_ijms, 2022, doi:10.3390/ijms231810610_

Round 1

Reviewer 1 Report

Review of

 Gene expression and chondrogenic potential of cartilage cells:

Osteoarthritis grades differences

Mazor et al.

General comments

·        I found the study straightforwardly designed and the research question useful.

·        The text could benefit from minor editing for grammar, spelling, and punctuation (see some examples below).

·        The primary revisions that are required are: (i) correcting the figure legends to match the corresponding figures, and (ii) clarifying the cell types being analyzed, and consistently stating which markers are associated with each cell type.

·        The Abstract convolutes changes over time with changes across cell types and grades of OA.

Specific comments

Title – “Osteoarthritis grade differences”

Line 16 – “mesenchymal progenitor cell (MPC) markers” (also line 54)

Line 29 – define MSC

Lines 46-47 – ACT, not ACI

Line 70 – “multipotent”; also lines 335 and 433.

Line 80 – Define OARSI

Line 85 – “placed in one”

Line 86 – Since the Methods section is at the end, the scoring system is described below, not above. Make sure all text is adjusted for this order of presentation (Results before Methods).

Line 88 – Define D0

Line 96 – Here CD105, CD166, and Sox9 are called MPC markers, but on lines 102-103, CD105 and CD166 are called MSC markers, while Sox9 is called a chondrocyte marker. Throughout the text, these types of convolutions caused me confusion, especially because the Introduction contrasted the properties of MPCs vs. MSCs in terms of their therapeutic benefit. Please revise the text for a consistent definition of cell types and their respective markers.

Line 100-105 – Figure legend does not match the figure. This seems to be true for most or all figures in the manuscript and requires major revision. In contrast, the figures themselves do seem to correspond to the descriptions in the text, so the problem appears to be limited to the legends only.

Line 117 – “live/dead ratio” (also line 109)

Line 132 – Do not refer to tables or figures in the section headings. Also lines 160-161, 178, and 204.

Line 135 – Here and in the Abstract, it was not clear what is meant by “the MSC cell line.” Was this a purchased cell line used as a "healthy control” comparator to the cells derived from the OA tissue? Above you say that MSCs are present in OA tissue, so I wasn’t sure whether these came from your tissue samples or from another source.

Line 142 – “moderate OA-derived cells”

Line 158 – In my opinion, Table 1 should be presented as supplementary data and not in the main manuscript.

Line 166 – Use consistent notation for day numbers.

Line 171 – Moderate or mild?

Line 176 – I think all the tables could be presented as supplementary information. The main story line seems to be presented thoroughly in the figures. See also lines 202 and 223.

Line 179 – “…that in mild…”

Line 188 – I believe this should refer to Figure 5, not Figure 4.

Line 190 – I believe this should refer to Figure 6, not Figure 4.

Lines 191-193 – Text referring to Figure 8 should appear in the appropriate section below, not here.

Line 226 – Define HES.

Lines 226-231 – There is no legend for Figure 8, and since mild, moderate, and severe samples appear in each panel, the text should refer to left, middle, and right images.

Lines 248-249 – Wouldn’t the mixture of grades confound your results? Please clarify.

Line 255 – Provide a reference for the OARSI grading system.

Line 263 – OARSI grade, or Outerbridge grade?

Line 275 – “…reached 80% confluence.”

Line 348 – If Aggrecan = Acan, this should be defined at first use.

Lines 346-350 – This seems to be the most accurate and concise description of the study. If it is indeed accurate, I suggest using it as a reference against which to compare all other descriptions of the cell types and their markers as you revise the text.

Line 358 – “…staining, which eliminates it…”

Line 386 – Spell out ECM.

Line 421 – Use the abbreviation defined in the Introduction: BM-MSC.

Line 432 – I agree with the general conclusions, but as noted above, some of the details need to be corrected and clarified

Author Response

We thank the reviewers for their comments and contribution to improve our manuscript. The manuscript has been revised according to the comments. Below is a point by point response to the reviewers’ comments (in blue) and all the corrections in manuscript are in the red pencil.

Reviewer 2 Report

The authors present a novel investigation of cartilage isolated from mild, moderate and severe OA human tissue.  Tissue was surgically isolated from 25 patients.  Tissue was isolated from tibial plateaus of individuals undergoing total joint replacement surgery.  Grafts were decalcified, paraffin embedded to characterize the severity of OA with OARSI scoring.  

The authors isolated cells for QPCR from with a scalpel.  The methods section is not clearly written.  It suggests that deparaffinized hisology sections were used for cell isolation.  Figure 1 shows that the cells were isolated from the tissue and expanded.  The authors should reword this to clarify.

Naiive cells and differentiated cells at day 35 were analyzed via QPCR.  Histology and immunostaining was completed for the expanded cells.

The major genes and proteins of interest were CD105, CD166, Notch 1 (MSC markers), Acan, and Col 2, and Sox9.

The results of this study critically demonstrate that there are residual functional stem cells present in degenerated articular cartilage.  The authors further show that these cells are functional and capable of producing articular cartilage.

This is a very important finding of the study and for the field as a whole. It may alter potential OA therapeudic strategies depending on the severity of OA.

Author Response

(The authors gave the same response as above.)

Reviewer 3 Report

Mazor et al., have reported that multipotential MPCs are present in human OA cartilage and they compared their chondrogenic potential by examining various markers including CD166, Notch1 as well as safranin O staining. Interestingly, they found that the re-differentiation state is different according to the severity of the arthritic tissue where they are taken. It is informative if we can select the suitable tissue for the regeneration therapy when we take the tissues during the surgery. It may result in improving therapeutic outcomes. However, there are many concerns to convince their findings.

1: They insist that moderate OA-derived cells are the best potential source for re-differentiation. However, there is no data for supporting their findings such as mechanism and reasons. Why “moderate” is the best, not ‘mild’? I think they cannot explain why. What is the underlying mechanism? Without the reason, it is not Science.

2: Among the 44 samples, the number of mild OA is small as compared with others.

3: The condition of OA cartilage is diverse and complex. For example, one may have a huge mechanical load if the patient is obese. Also, the medical history is not uniform including the therapeutic record. 

4: There are many disagreements between figures and figure legends (e.g, Fig.1 and Fig.1 legend). It is embarrassing that they did not check carefully before the submission. Moreover, in the Fig. 1 legend, they described Acan, Col II, and Col I but they don’t appear in Fig. 2 (I guess favorably Fig. 2 was originally Fig. 1 when this legend was prepared).

5: Figure2: The cell number and viability data should not be indicated by mean+/-standard error because the data are not normally distributed. If they want to compare the groups, they may use the median.

6: As they showed in Fig.2, a lot of dead cells are included. Did they eliminate the dead cell during the following experiments? I can see a lot of Notch 1 negative cells (= 0.0) even in the moderate derived cells.

7: Fig.7: What is the box in Fig. 7B and 7C? There is no explanation in the figure legend and the Results section.

8: Overall, they suggested that moderate OA-derived cells are best for cell-based regenerative therapy. However, there is little chance to obtain ‘moderate’ tissues in OA surgery. That is the crucial point for thinking of its potential as a source for cell-based regenerative therapy.

Minor: Page 2 line 54: “Mesenchymal progenitor cell markers (MPC)” should be “Mesenchymal progenitor cells (MPC)”.

Author Response

(The authors gave the same response as above.)

Round 2

Reviewer 1 Report

The following items still need to be addressed:

1. Many grammatical errors remain. The text needs to be edited for English. For example, the Abstract sentence from line 32 to 34 is missing a verb and an object.

2. Line 93 - D0 has not been defined.

3. Abstract lines 21-23 need to be edited for punctuation to accurately define the three groups of markers. Per my understanding of the groupings, Fig. 5-7 legends are incorrect in their classification of Sox9.

4. Line 179 - Sox9 is called a marker of chondrogenesis, but the Abstract defines Sox9 as an MPC marker.

5. Line 128 - the originally requested change was not made (use "live/dead", not "live/death").

6. Lines 204 and 206 still refer to Fig. 4, which the authors agreed was not correct.

Author Response

Dear Editor, we thank the reviewers for their comments and contribution to improve our manuscript. The manuscript has been revised according to the comments. Below is a point by point response to the reviewers’ comments (in blue) and all the corrections in manuscript are in the red pencil.

Reviewer 1

Comments and Suggestions for Authors

The following items still need to be addressed:

  1. Many grammatical errors remain. The text needs to be edited for English. For example, the Abstract sentence from line 32 to 34 is missing a verb and an object.

Answer: Thank you, the manuscript has been revised grammatically by an English native.

  1. Line 93 - D0 has not been defined.

Answer: It has been corrected.

  1. Abstract lines 21-23 need to be edited for punctuation to accurately define the three groups of markers. Per my understanding of the groupings, Fig. 5-7 legends are incorrect in their classification of Sox9.

Answer: Thank you, the punctuation and figures legends were corrected.

  1. Line 179 - Sox9 is called a marker of chondrogenesis, but the Abstract defines Sox9 as an MPC marker.

Answer: Thank you, corrected.

  1. Line 128 - the originally requested change was not made (use "live/dead", not "live/death").

Answer: Thank you, corrected.

  1. Lines 204 and 206 still refer to Fig. 4, which the authors agreed was not correct.

Answer: correccted.

Reviewer 3 Report

This reviewer requested that they should use median, not mean, when the data are not normally distributed. In the response letter, the authors answered “We agree. Data has been corrected by using median.” However, in line 371, they again stated that “Data are reported as mean…”.

Author Response

Dear Editor, we thank the reviewers for their comments and contribution to improve our manuscript. The manuscript has been revised according to the comments. Below is a point by point response to the reviewers’ comments (in blue) and all the corrections in manuscript are in the red pencil.

Reviewer 3

Comments and Suggestions for Authors

This reviewer requested that they should use median, not mean, when the data are not normally distributed. In the response letter, the authors answered “We agree. Data has been corrected by using median.” However, in line 371, they again stated that “Data are reported as mean…”.

Answer: Thank you, we agree. We have added a sentence in the text to explain that in figure 3. we reported our data as a median.

Round 3

Reviewer 1 Report

The revisions adequately addressed my comments.

Author Response

Reviewer 1

Comments and Suggestions for Authors

English language and style are fine/minor spell check required

Answer: Thank you, the manuscript has been revised by an English native speaker.

Reviewer 3 Report

The authors wrote "Data are reported as mean ± SEM in all the figure except figure 3."

Please check that whether all the data shown in this manuscript were normally distributed or not. I requested that authors use "mean" when the data were normally distributed (e.g., Fig 8). When the data were not normally distributed as seen in Fig. 2, 3, 4, (5?), 6, and 7, authors should use median ± SEM.

Author Response

Reviewer 3.

Comments and Suggestions for Authors

The authors wrote "Data are reported as mean ± SEM in all the figure except figure 3."

Please check that whether all the data shown in this manuscript were normally distributed or not. I requested that authors use "mean" when the data were normally distributed (e.g., Fig 8). When the data were not normally distributed as seen in Fig. 2, 3, 4, (5?), 6, and 7, authors should use median ± SEM.

We thank the reviewer for his comment

We agree the data were not normally distributed. Therefore, we have applied  a Non – parametric Kruskal Wallis and Mann Whitney test> We have also  modified the figures and expressed median with 95% confidence interval.

Round 4

Reviewer 3 Report

no comments